# Linear ubiquitination regulates the KSHV replication and transcription activator protein to control infection

Yi Luan [1,2,3,4,10], Wenying Long[5,10], Lisi Dai[6,7,8], Panfeng Tao[9], Zhifen Deng[1,2,3,4] & Zongping Xia [1,2,3,4] ✉

Like many other viruses, KSHV has two life cycle modes: the latent phase and the lytic phase. The RTA protein from KSHV is essential for lytic reactivation, but how this protein's activity is regulated is not fully understood. Here, we report that linear ubiquitination regulates the activity of RTA during KSHV lytic reactivation and de novo infection. Overexpressing OTULIN inhibits KSHV lytic reactivation, whereas knocking down OTULIN or overexpressing HOIP enhances it. Intriguingly, we found that RTA is linearly polyubiquitinated by HOIP at K516 and K518, and these modifications control the RTA's nuclear localization. OTULIN removes linear polyubiquitin chains from cytoplasmic RTA, preventing its nuclear import. The RTA orthologs encoded by the EB and MHV68 viruses are also linearly polyubiquitinated and regulated by OTULIN. Our study establishes that linear polyubiquitination plays a critically regulatory role in herpesvirus infection, adding virus infection to the list of biological processes known to be controlled by linear polyubiquitination.

Kaposi's sarcoma-associated herpesvirus (KSHV), or human herpesvirus 8 (HHV8), is a member of the Gamma herpesvirus subfamily. This oncogenic virus is linked with three types of malignancies: Kaposi's sarcoma (KS)[1,2], primary effusion lymphoma (PEL)[3], and multicentric Castleman's disease (MCD)[4]. Similar to other herpesviruses, KSHV has a life cycle mode of two phases: latent and lytic[5]. KSHV mainly maintains in the latent phase that is characterized by a specific profile of gene expression. However, it can be lytically reactivated under stressors such as hypoxia or exposure to chemical agents like 12-O-tetra-decanoylphorbol-13-acetate (TPA), sodium butyrate (NaB), and valproate (VPA)[6-8], which is characterized by the sequential expression of lytic genes, including immediate-early (IE), early (E), and late (L)

genes[9,10]. KSHV lytic reactivation is crucial for both viral propagation and KSHV-induced tumors, which provide therapeutic opportunities for KSHV-related malignancies.

KSHV lytic reactivation from latency is governed by a protein encoded by its ORF50, the replication and transcription activator (RTA). Studies have demonstrated that RTA expression is required to initiate KSHV transition to the lytic phase from its latency[11,12]. RTA is highly conserved in gamma-herpesvirus family (e.g., Epstein-Barr virus (EBV) and murine herpesvirus 68 (MHV68))[13]. RTA has arginine (R)- and lysine (K)-rich NLS (nucleus localization signal) motifs at both its N-terminus (NLS1, aa 6-12) and C-terminus (NLS2, aa 514-528), and the NLS2 is essential for its nuclear localization[14]. As an IE gene, RTA acts as

[1]Clinical Systems Biology Laboratories, Translational Medicine Center, The First Affiliated Hospital of Zhengzhou University, Zhengzhou, Henan, China. [2]Institute of Infection and Immunity, Henan Academy of Innovations in Medical Science, Zhengzhou, China. [3]Department of Neurology, the First Affiliated Hospital of Zhengzhou University, Zhengzhou, Henan, China. [4]NHC Key Laboratory of Prevention and Treatment of Cerebrovascular Diseases, the First Affiliated Hospital of Zhengzhou University, Zhengzhou, Henan, China. [5]Center for Clinical Research, the Fourth Affiliated Hospital of School of Medicine, and International School of Medicine, International Institutes of Medicine, Zhejiang University, Yiwu, Zhejiang, China. [6]Department of Pathology & Pathophysiology of Second Affiliated Hospital, Zhejiang University School of Medicine, Hangzhou, Zhejiang, China. [7]Department of Surgical Oncology of Second Affiliated Hospital, Zhejiang University School of Medicine, Hangzhou, Zhejiang, China. [8]School of Basic Medical Sciences, Zhejiang University, Hangzhou, Zhejiang, China. [9]Life Sciences Institute, Zhejiang University, Hangzhou, Zhejiang, China. [10]These authors contributed equally: Yi Luan, Wenying Long. ✉e-mail: zxia2018@zzu.edu.cn

a transcriptional factor to promote the sequential expression of lytic genes through its interaction with RTA responsive elements (RREs), resulting in the generation of viable viral particles. RTA can drive the expression of multiple viral genes, including PAN RNA, Ori-Lyt-associated RNA, ORF57, K8, and RTA itself[15]. RTA also promotes ubiquitin-proteasome-mediated degradation of RTA repressors[16,17]. In addition, RTA functions as a ubiquitin E3 ligase that targets MyD88 and IRF7 for ubiquitination and subsequent degradation, thereby enabling KSHV to evade immune attacks from host cells[12,18,19]. Interestingly, this E3 ligase activity can catalyze RTA self-ubiquitination, thereby maintaining RTA levels relatively low to restrict unwanted viral reactivation[19,20]. Notably, the transcriptional function of RTA is negatively regulated by multiple post-translational modifications (PTMs), which include PARP-1-mediated poly(ADP-ribosylation), the kinase hKFC-mediated phosphorylation[21], and O-GlcNAcylation facilitated by O-GlcNAc transferase (OGT) at its Ser/Thr-rich motifs[22]. Despite these progresses made about the regulation of RTA activity, the mechanisms by which its activity is regulated are not fully understood.

Protein ubiquitination, a PTM essential for many cellular processes including protein degradation, DNA damage responses, protein trafficking, and intracellular signaling, involves the conjugation of different types of ubiquitin chains to lysine residues on target proteins[23]. The known ubiquitin chains consist of the seven canonical lysine-linked polymer chains and the more-recently discovered linear Met1-linked polyubiquitin (M1 polyUb) chains[24]. M1 ubiquitination has been revealed to be important in signaling activation pathways, such as NF-κB and RIG-I in immune and inflammatory signaling processes. It is also known to regulate cell death, T- and B-cell development, murine embryonic development, cancer, and autoimmune diseases[25–27]. M1 polyUb chains are formed when the amino terminus of one ubiquitin is conjugated to the carboxyl terminus of another ubiquitin[24]. The E3 ligase "linear Ub chain assembly complex (LUBAC)" is currently the only discovered ligase capable of generating M1 polyUb chains. LUBAC consists of three subunits, HOIP, HOIL-1L, and Sharpin, of which HOIP is responsible for the ligase activity[24,28,29]. Additionally, cells encode a deubiquitination (DUB) enzyme OTULIN (Ovarian TUmor deubiquitination enzyme with LINear linkage specificity; also called FAM105B and Gumby) that specifically antagonizes the M1 polyUb chains[30,31]. The opposing activities of LUBAC and OTULIN together determine the overall abundance of M1 polyUb chains in cells. Studies are revealing the involvement of M1 polyUb chains in many biological processes. Notably, M1 polyUb chains have been reported in regulating HTLV Tax function[32].

In this study, we investigated the possible roles of ubiquitination in controlling KSHV lytic reactivation. We found that M1 ubiquitination regulates RTA during KSHV infection. Specifically, overexpression of OTULIN inhibited KSHV lytic reactivation, while OTULIN knockdown and HOIP overexpression each enhanced it in iSLK/rKSHV.219 cells. RTA was M1 polyubiquitinated by HOIP at K516 and K518, and these modifications controlled the localization of RTA in the nucleus. We showed that OTULIN can remove M1 polyubiquitin chains from RTA, thereby enabling the relocation of RTA from the nucleus to the cytoplasm. We also revealed that the RTA orthologs encoded by the EBV and MHV68 viruses are also M1 polyubiquitinated and are regulated by OTULIN. In addition to addressing the biological functions of M1 polyubiquitination in regulating virus infection, our research offers valuable perspectives on potential targets of therapeutics for combating KSHV infection and KSHV-associated malignancies.

## Results

### OTULIN suppresses KSHV lytic reactivation
In the investigation of cellular mechanisms that might regulate KSHV lytic reactivation, we initially directed our attention to the DUB enzyme family. We utilized the iSLK/rKSHV.219 cell line, which contains a recombinant KSHV (rKSHV.219) genome featuring constitutive GFP expression to indicate latent infection and induced RFP expression upon lytic cycle induction to signify lytic reactivation[33]. Following preliminary experiments, our focus turned to OTULIN, a DUB enzyme known for its specificity towards M1 polyUb chains. To examine the effect of OTULIN on KSHV lytic reactivation, we transfected iSLK/rKSHV.219 cells with an empty vector, wild type (WT) OTULIN, or its deubiquitinating activity-deficient mutant OTULIN[C129A][31] (Supplementary Fig. 1a). Subsequent induction of lytic reactivation with tetracycline (Tet) and valproate (VPA) revealed a notable increase in RFP+ cells in those transfected with the empty vector or OTULIN[C129A], whereas cells transfected with OTULIN exhibited a significant decrease in RFP+ cells (Fig. 1a and b). Analysis of viral particles in the culture media demonstrated that cells transfected with OTULIN released a reduced number of viral particles compared to those transfected with empty vector or OTULIN[C129A] (Supplementary Fig. 1b). Similarly, the media from OTULIN-transfected cells contained fewer infectious viral particles compared to media from cells transfected with the empty vector or OTULIN[C129A] (Fig. 1c).

We also employed siRNAs to downregulate OTULIN expression levels in iSLK/rKSHV.219 cells[30,34]. The efficiency of OTULIN knockdown (KD) was verified by RT-qPCR and Western blotting (Supplementary Fig. 1c, d). Photomicrographs and flow cytometry analysis showed that the introduction of three distinct siRNA pairs targeting OTULIN led to substantial increases in the percentage of RFP+ cells when compared to cells treated with control siRNA (Fig. 1d, e). Furthermore, knockdown of OTULIN led to an elevation of viral particles in the culture media (Fig. 1f). These results indicate that OTULIN inhibits KSHV lytic reactivation in a deubiquitinase-activity-dependent manner.

### M1 polyubiquitination regulates RTA transcriptional activity
We investigated whether OTULIN's inhibitory effect on KSHV lytic reactivation stems from its ability to suppress the expression of lytic genes. RT-qPCR analysis revealed that, in comparison with empty vector or OTULIN[C129A] transfection, OTULIN transfection led to a significant reduction in the mRNA levels of key KSHV lytic genes, including *RTA, ORF59, ORF9,* and *ORF8.1* in iSLK/rKSHV.219 cells (Fig. 2a). Conversely, OTULIN KD led to a dramatic enhancement in the expression of these lytic genes (Fig. 2b).

Given that RTA serves as the established trigger for KSHV lytic activation[35], we performed reporter assays to investigate whether OTULIN directly regulates RTA's transcriptional activity. While RTA expression enhanced the luciferase activities of pGL3-K8-DE250-Luc and pGL4-RTA-Luc reporters, co-expression with OTULIN attenuated their activities (Supplementary Fig. 2a, b). Additionally, co-expression with OTULIN, but not with OTULIN[C129A] dose-dependently repressed RTA transcriptional activity in the pOri-Lyt-Luc reporter assay (Fig. 2c). Notably, co-expression of RTA with DUBs IsoT, CYLD, or the viral OUT domain-containing DUB (vOTU) had no effects on RTA's transcriptional activity (Supplementary Fig. 2c). Interestingly, co-expression with the E3 ligase HOIP, but not its ligase activity-deficient mutant HOIP[CS], which can only form linear ubiquitin chains, enhanced RTA transcriptional activity (Supplementary Fig. 2d).

To investigate how OTULIN inhibits RTA's transcriptional activity, we conducted ChIP assays in both HEK293T cells and iSLK/rKSHV.219 cells. In HEK293T cells transfected with RTA and either OTULIN, HOIP, or OTULIN[C129A] together with the reporter plasmids pOri-Lyt-Luc or pGL4-RTA-Luc, OTULIN expression reduced the binding of RTA with *Ori-lyt* (Fig. 2d) and *RTA* promoters (Supplementary Fig. 2e), while HOIP increased RTA's interaction with both promoters, and OTULIN[C129A] had no significant effect. Similarly, in lytically reactivated iSLK/rKSHV.219 cells, OTULIN expression diminished, while HOIP enhanced the interaction of endogenous RTA with both *PAN* (Fig. 2e) and *ORF57* (Fig. 2f) promoters. Collectively, these findings indicate that M1 ubiquitination regulates the

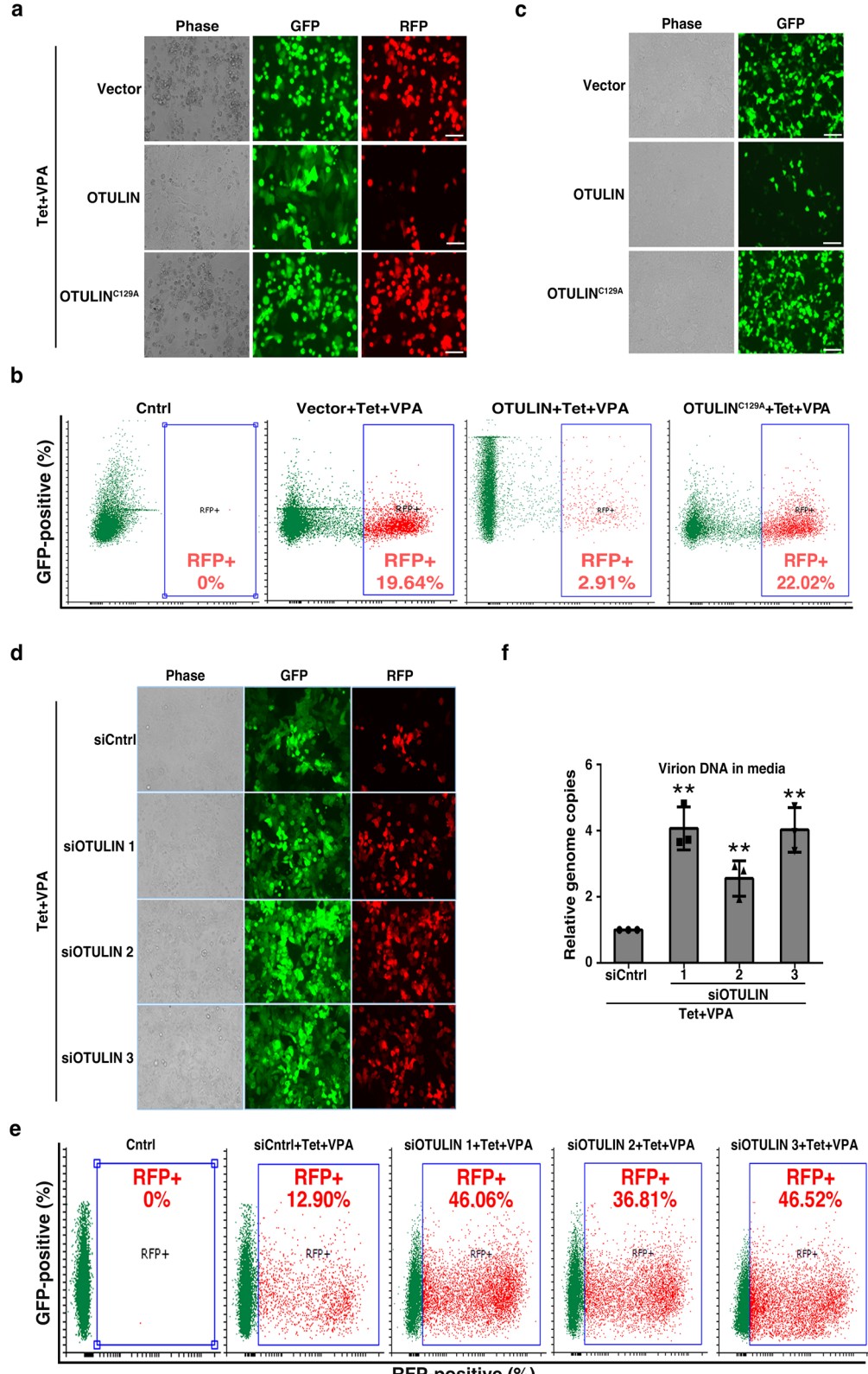

transcriptional activities of RTA by modulating its interaction with target gene promoters.

### RTA K516 and K518 can be modified via M1 ubiquitination

Given that M1 polyubiquitination regulates RTA transcriptional activity, we next sought to determine whether RTA itself undergoes M1 polyubiquitination. We transfected HEK293T cells with RTA-Flag together with or without Ub[KO], a ubiquitin mutant with all of its lysine residues being mutated to arginine residues and forming M1 polyUb chains only, for an intracellular ubiquitination assay[36]. Under denaturing conditions, we immunoprecipitated RTA-Flag and performed immunoblotting using a specific anti-linear Ub antibody. Polyubiquitinated RTA was detected by this antibody, and its levels were enhanced in the presence of Ub[KO] (Fig. 3a). In addition, co-

**Fig. 1 | The linear deubiquitination enzyme OTULIN suppresses KSHV lytic reactivation. a**–**c** Overexpression of OTULIN inhibits KSHV lytic reactivation. iSLK/rKSHV.219 cells were transfected with vector, OTULIN, or OTULIN$^{C129A}$ for one day followed by treatment with tetracycline (Tet) plus valproate (VPA) for two days to induce KSHV lytic reactivation. The cells were photographed for GFP and RFP fluorescence (**a**), or collected for flow cytometric evaluation of RFP fluorescence (**b**). The gating strategy is shown in Supplementary Fig. 6a. The culture media were also collected and used to infect HEK293T cells. Photomicrographs showing phase and GFP fluorescence were taken two days post infection (**c**). Scale bar: 250 μm. Image is representative in total n = 3 independent experiments. **d**–**f** Knockdown of OTULIN enhances KSHV lytic reactivation. iSLK/rKSHV.219 cells were transfected

with control siRNA (siCntrl) or three different pairs of siRNAs targeting OTULIN (siOTULIN 1, 2 and 3). After three days, the cells were treated with Tet and VPA for two more days and photographs were taken (**d**) or collected for flow cytometric evaluation of RFP fluorescence (**e**). The gating strategy is shown in Supplementary Fig. 6a. Image is representative in total n = 3 independent experiments. Culture media were also collected for DNA extraction. Viral DNA in the media was quantified by qPCR. Data are presented as means of three technical replicates (**f**), n = 3 biological replicates and 2 technical replicates, mean ± s.d., two-sided t test, ** P = 0.00122, 0.00729, 0.00149 (siCntrl vs. siOTULIN1, 2, 3, respectively). Source data are provided as a Source Data file.

expression of HOIP and HOIL-1L also resulted in enhancement of RTA polyubiquitination (Fig. 3b). To corroborate these results, we performed intracellular deubiquitination assays in HEK293T and iSLK/rKSHV.219 cells. Expression of OTULIN, but not OTULIN$^{C129A}$, led to a significant reduction in the amount of polyubiquitinated RTA (Fig. 3c, d).

Next, seeking to identify the M1 polyubiquitination sites on RTA, we initially speculated the sites might be in the RTA NLS2 motif and generated RTA mutants with K to R mutations in the NLS2 motif to mimic their non-ubiquitinated states (Fig. 3e). Testing of their transcriptional activation activities in the luciferase reporter assay showed that among the six single site mutants, only RTA$^{K516R}$ and RTA$^{K518R}$ displayed a modest reduction in their transcriptional activities, with OTULIN further inhibiting their activities (Supplementary Fig. 3a). These results suggest that K516 and K518 of RTA are M1 polyubiquitination sites. Consistent with this assertion, the double-site mutant RTA$^{2KR}$ (combination of K516R and K518R) displayed a dramatic decrease in its transcriptional activity, and OTULIN no longer inhibited its activity (Supplementary Fig. 3b). As a control, the quadruple-site mutant RTA$^{4KR}$ (combination of K522R, K527R, K528R, and K530R) displayed an activity as that of WT RTA (Supplementary Fig. 3b). Furthermore, a ubiquitination assay showed that RTA$^{2KR}$ had greatly reduced M1 polyubiquitination levels as compared to WT RTA; the presence of OTULIN reduced the M1 polyubiquitination level of WT RTA but not that of RTA$^{2KR}$ (Fig. 3f).

## M1 polyubiquitination regulates RTA subcellular localization

Having confirmed that RTA is modified with M1 polyUb chains, we next examined how this modification may affect RTA's transcriptional activity. As RTA is a transcription factor, we investigated whether this M1 polyubiquitination modification may influence its subcellular localization. Upon separation of cellular extracts into cytoplasmic and nuclear fractions, we found that the RTA immunoprecipitated from the nuclear fraction had a strong signal for M1 ubiquitination, whereas RTA from the cytoplasmic fraction only displayed a very weak signal, implying M1 polyubiquitination might regulate the cytoplasm vs. nucleus distribution of RTA (Fig. 4a).

Further supporting a direct role in regulating subcellular localization, co-immunoprecipitation revealed that OTULIN physically interacts with RTA in both HEK293T co-expressing the two proteins and in iSLK/rKSHV.219 cells upon lytic reactivation (Supplementary Fig. 4a–c). We also conducted subcellular fractionation assays in HeLa and iSLK/rKSHV.219 cells to determine if OTULIN affects the subcellular localization of RTA. Western blotting revealed that RTA was predominantly detected in the nuclear fractions. However, upon expression of OTULIN, but not OTULIN$^{C129A}$, RTA was observed in both the cytoplasmic and the nuclear fractions (Fig. 4b, c). Immunofluorescence analysis also revealed the same pattern: RTA was mainly localized in the nucleus but was redistributed to the cytoplasm upon expression of OTULIN (but not OTULIN$^{C129A}$) in HEK293T cells (Fig. 4d, e). We also examined the subcellular localization of RTA$^{2KR}$ and RTA$^{4KR}$ mutants. RTA$^{2KR}$ was detected in both the cytoplasmic and nuclear compartments and

neither OTULIN nor OTULIN$^{C129A}$ had any effect on its subcellular localization in both HeLa cells (Fig. 4f) and HEK293T cells (Fig. 4g), whereas RTA$^{4KR}$ was localized in the nucleus but redistributed to the cytoplasm upon expression of OTULIN (but not OTULIN$^{C129A}$) (Supplementary Fig. 4d, e).

Collectively, these results demonstrate that nuclear but not cytoplasmic RTA is modified with M1 polyubiquitination and establish that OTULIN facilitates the relocation of RTA from the nucleus to the cytoplasm.

## OTULIN regulates KSHV lytic reactivation

To explore the potential role of OTULIN in lytic reactivation in other KSHV-harboring cells, we transfected siRNAs targeting OTULIN into BCBL-1 cells to significantly reduce OTULIN expression as evaluated by RT-qPCR, immunoblotting and immunofluorescence (Fig. 5a, f, h). BCBL-1 is a B cell lymphoma cell line generated from the peritoneal effusion of a primary effusion lymphoma patient, harboring latently-infected KSHV[37]. The cells were subjected to NaB and TPA treatment to initiate KSHV lytic induction, which resulted in the increases of mRNA levels of the lytic genes *RTA*, *ORF59*, *ORF9*, and *ORF8.1* in OTULIN KD cells (Fig. 5b). This increased lytic gene expression was associated with an increase in viral particle secretion into the culture media (Fig. 5c). Meanwhile, in these reactivated lytic BCBL-1 cells, OTULIN knockdown enhanced the interaction of endogenous RTA with the *ORF57* (Fig. 5d) and *ORF59* promoters (Fig. 5e). We also performed subcellular fractionation assays to determine if OTULIN knockdown affects the subcellular localization of RTA. Immunoblotting revealed that RTA levels were increased upon TPA plus NaB induction and were further elevated in the nuclear fractions upon OTULIN knockdown (Fig. 5f). Immunofluorescence analysis also revealed the same pattern: RTA was predominantly localized in the nucleus and was further enhanced in the nucleus of OTULIN KD cells when induced with TPA and NaB (Fig. 5g).

We surmised that OTULIN might also influence KSHV de novo infection. To verify this speculation, we utilized shRNAs targeting OTULIN to establish stable OTULIN KD cell lines with substantial reduction of OTULIN in HUVEC cells (Supplementary Fig. 5f). Subsequently, we infected these cells with rKSHV.219. These cells showed a dramatic increase in GFP$^+$ cells when compared to control shRNA-treated cells (Supplementary Fig. 5a, b). Moreover, this increase was in parallel with an increase in mRNA levels of KSHV genes *vFlip*, *vCyclin*, and *RTA* (Supplementary Fig. 5c). Interestingly, after generating latently infected cells and treating them with CoCl$_2$ to induce lytic reactivation, we found that HUVEC cells with OTULIN depletion displayed significant increases in the proportion of RFP$^+$ cells and secreted more viral particles into the cell culture media than cells generated from control shRNA (Supplementary Fig. 5d and e). CoCl$_2$ treatment can stabilize the protein level of the hypoxia-responsive transcription factor HIF1α in cells and thereby mimic a hypoxic environment[38]. These results collectively show that OTULIN inhibits KSHV lytic reactivation in both BCBL-1 and HUVEC cells, and de novo KSHV infection in HUVEC cells.

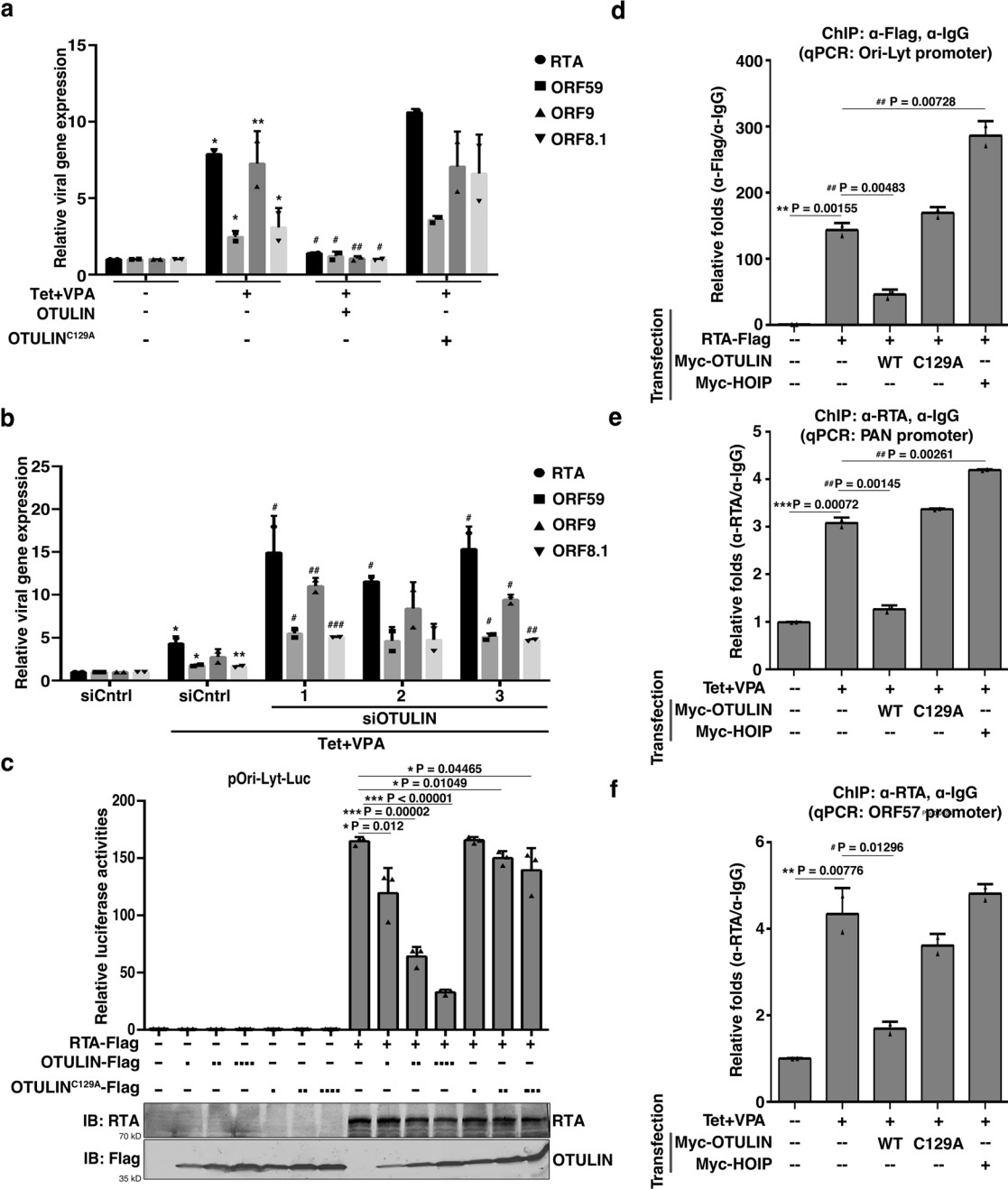

**Fig. 2 | Linear ubiquitination machinery regulates the transcriptional activity of KSHV RTA by modulating its binding to target gene promoters. a, b** iSLK/rKSHV.219 cells with OTULIN overexpression (**a**) or knockdown (**b**) were treated as indicated. RT-qPCR was performed to quantify the expression levels of KSHV lytic genes *RTA*, *ORF59*, *ORF9*, and *ORF8.1*. a, *n* = 2 biological replicates and 2 technical replicates, * P = 0.02288, 0.03410, 0.00229, 0.02574 (Cntrl vs. Tet/VPA) for *RTA*, *ORF59*, *ORF9*, *ORF8.1*; #, P = 0.02796, 0.03488, 0.00233, 0.03357 (OTULIN+Tet/VPA vs. Tet/VPA) for *RTA, ORF59, ORF9, ORF8.1*. **b**, *n* = 2 biological replicates and 2 technical replicates, * P = 0.03772, 0.03042, 0.00704 (siCntrl vs. siCntrl+Tet/VPA) for *RTA, ORF59, ORF8.1*; #, P = (0.03734, 0.02794, 0.03573), (0.03299, 0.02145), (0.00646, 0.01030), (0.00027, 0.00276) (siCntrl vs. siOTULIN) for *RTA, ORF59, ORF9, ORF8.1*. **c** OTULIN inhibits RTA transcriptional activities in reporter assays. HEK293T cells were co-transfected with luciferase reporter plasmid pOri-Lyt-Luc, a Renilla luciferase control plasmid, together with or without RTA, in the presence of increased amount of OTULIN or OTULIN^C129A, then harvested and their luciferase

activities were quantified. *n* = 3 biological replicates and 2 technical replicates. **d** OTULIN inhibits, while HOIP enhances, RTA binding to the *Ori-Lyt* promoter. HEK293T cells were co-transfected with the reporter plasmid pOri-Lyt-Luc, RTA-Flag, and with Myc-OTULIN, Myc-HOIP, or Myc-OTULIN^C129A as indicated. Cells were collected for ChIP assay using anti-Flag magnetic agarose beads. Precipitated *Ori-Lyt* promoter DNAs were quantified using qPCR. *n* = 2 biological replicates and 2 technical replicates. **e, f** OTULIN inhibits, while HOIP enhances, RTA binding to its target gene promoters. iSLK/rKSHV.219 cells were transfected with Myc-OTULIN, Myc-OTULIN^C129A, or Myc-HOIP followed by treatment with Tet plus VPA. The cells were collected and subjected to ChIP assay using an anti-RTA antibody. The quantities of *PAN* (**e**) or *ORF57* (**f**) promoter DNA in the precipitates were determined by qPCR, **e, f**, *n* = 2 biological replicates and 2 technical replicates. Data are displayed as mean ± s.d., two-sided t test. Source data are provided as a Source Data file.

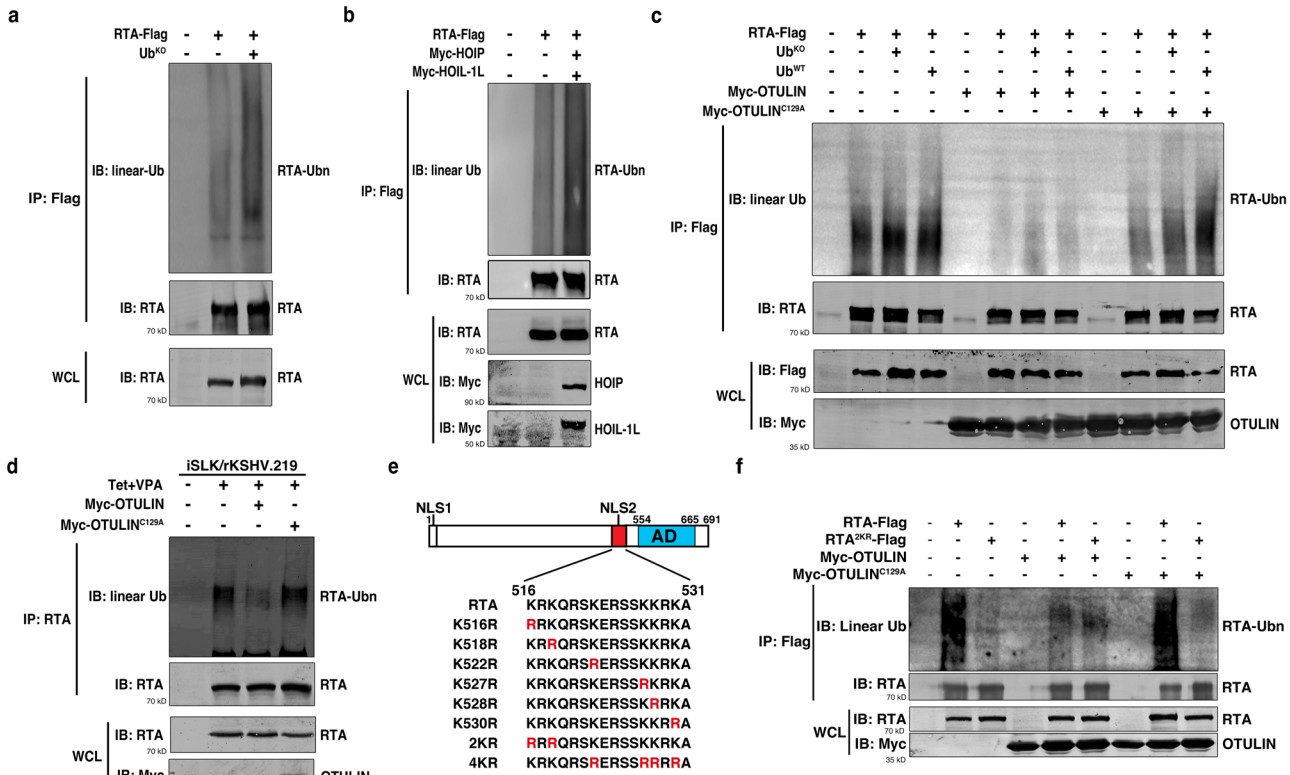

**Fig. 3 | HOIP/OTULIN control the M1 ubiquitination of RTA-K516 and -K518.**
**a** RTA is modified by M1 ubiquitination. RTA-Flag and Ub[KO] were co-transfected into HEK293T cells and subjected to immunoprecipitation with anti-Flag magnetic beads. RTA M1 ubiquitination was determined by a specific anti-linear Ub antibody (Genentech). Ub[KO] is a ubiquitin mutant that can only form M1 polyUb chains. $n = 3$ independent experiments. **b** HOIP/HOIL-1L promotes RTA M1 ubiquitination. RTA-Flag and Myc-HOIP/Myc-HOIL-1L were co-transfected into HEK293T cells. RTA-Flag was immunoprecipitated with anti-Flag magnetic beads and detected with anti-linear Ub and RTA antibodies, respectively. $n = 3$ independent experiments. **c** OTULIN reduces RTA M1 ubiquitination. As in (**b**), but the plasmids were replaced with the indicated plasmids. $n = 3$ independent experiments. **d** Endogenous RTA displays M1 ubiquitination. iSLK/rKSHV.219 cells were transfected with Myc-OTULIN or Myc-OTULIN[C129A] and then treated with Tet plus VPA. RTA was immunoprecipitated with anti-RTA antibody and immunoblotted with anti-linear Ub and anti-RTA antibodies, respectively. $n = 3$ independent experiments. **e** Diagrams of RTA domains, the amino acid sequence of the RTA NLS2 and the Lysine (K) to Arginine (R) mutants used in the study. **f** RTA[2KR] is defective in M1 ubiquitination. HEK293T cells were transfected with plasmids as indicated and cell lysates were immunoprecipitated using anti-Flag beads. Immunoprecipitated proteins were immunoblotted with the indicated antibodies. $n = 3$ independent experiments. Source data are provided as a Source Data file.

## OTULIN regulates RTAs encoded by other gamma-herpes viruses

To evaluate the effect of OTULIN on the activity of RTAs encoded by other gamma-herpes viruses, we tested the RTA orthologs from EBV and MHV68, two other representative gamma-herpes viruses. Both EBV RTA and MHV68 RTA augmented the activation of the pOri-Lyt-Luc reporter, and their activities were suppressed upon co-expression with OTULIN but not OTULIN[C129A] (Fig. 6a). In addition, similar to KSHV RTA, both EBV RTA and MHV68 RTA were modified by M1 polyubiquitination, and co-expression with OTULIN but not OTULIN[C129A] reduced this modification (Fig. 6b, c). Thus, KSHV RTA, EBV RTA, and MHV68 RTA are each modified by linear ubiquitination, and OTULIN inhibits their transcriptional activities.

## Discussion

With all the findings reported here, we propose a model for how M1 polyubiquitination regulates KSHV lytic reactivation (Fig. 6d). LUBAC specifically ubiquitinates RTA on K516 and K518, which enables RTA translocation into nucleus where it can activate the expression of a lytic gene cascade that initiates KSHV lytic reactivation. Opposing this process, OTULIN cleaves the linear ubiquitination modification from RTA and thereby inhibits its nuclear translocation, thus inhibiting KSHV lytic reactivation. There are triggering events (*e.g.*, hypoxia, HIV coinfection, among others) that initiate KSHV lytic reactivation; but

the molecular mechanisms that connect KSHV lytic reactivation and the triggering events are not fully understood. We reported here that under hypoxic condition the M1 polyubiquitination machinery is involved in RTA regulation during KSHV lytic reactivation (Supplementary Fig. 5d), implying M1 polyubiquitination may be an important mechanism that connects KSHV lytic reactivation and its triggering events. It will be very interesting to investigate whether the enzymatic activities of LUBAC and OTULIN can be modulated by triggering events during KSHV lytic reactivation. It will also be intriguing to explore whether KSHV has mechanism(s) that can modulate M1 ubiquitination machinery as it infects host cells.

RTA has two putative NLS motifs, and it has been established that the NLS2 is required and sufficient for RTA nuclear localization[15]. The NLS2 is a classical nuclear localization motif and is characterized by clusters of basic residues K and R. These residues mediate RTA binding to importins α and β through their positive charge for RTA nuclear translocation. This has been recognized as the major regulation step in controlling RTA's nuclear translocation[39]. Our finding that M1 ubiquitination regulates RTA nuclear translocation introduces an additional level of regulation controlling RTA's nuclear translocation. It is not clear yet how M1 polyubiquitination specifically regulates RTA nuclear translocation, but there are several reasonable possibilities. M1 polyubiquitination might facilitate RTA's binding to importin α/β. Alternatively, it is conceivable that the M1 polyubiquitin chains do not affect

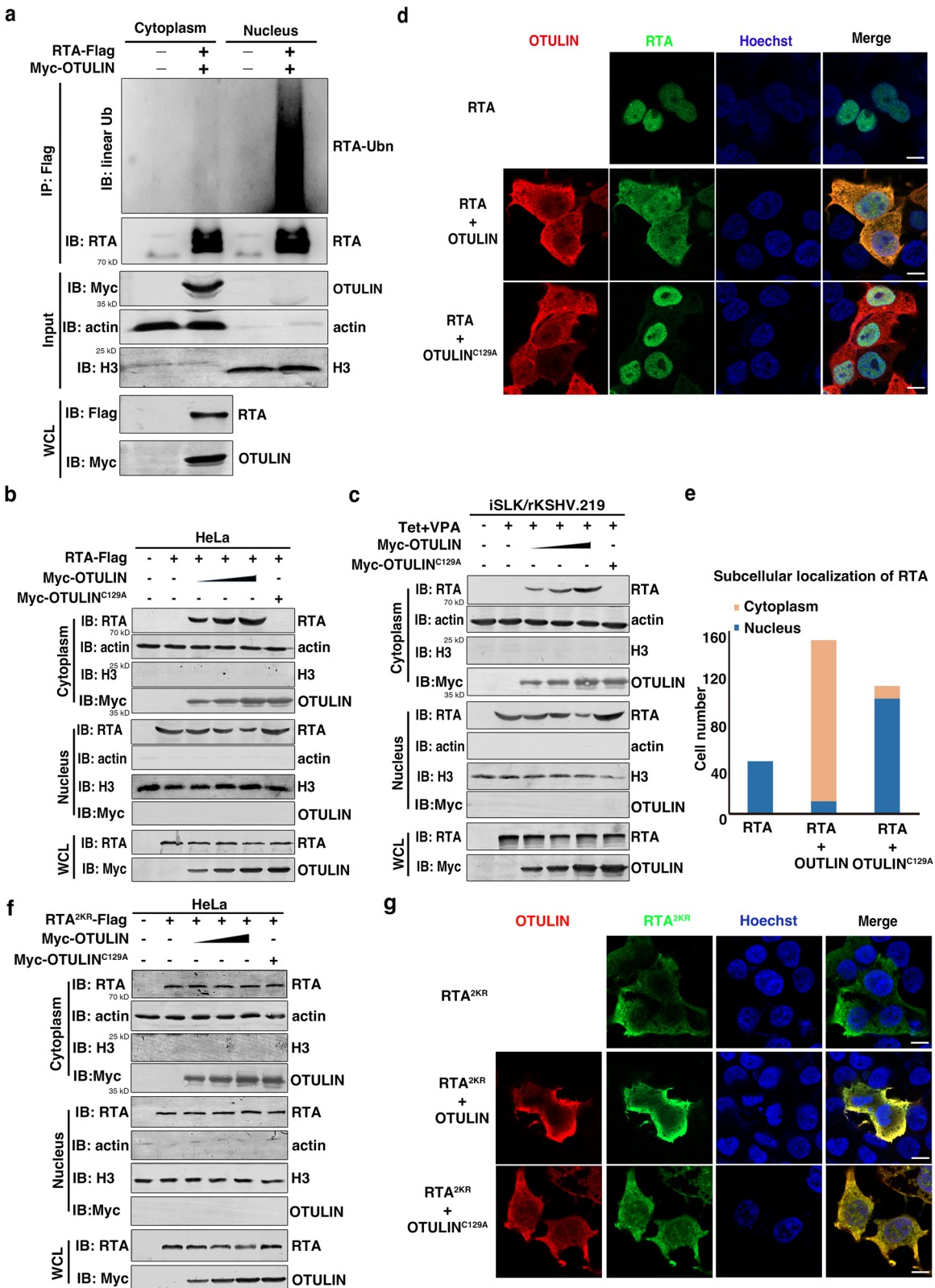

RTA nuclear translocation, but may instead function to retain RTA in the nucleus, by facilitating its binding to other factors in the nucleus (*e.g.*, transcriptional co-activators, target gene promoters, etc.) or via some other mechanism(s) like facilitation of RTA oligomerization into active tetramers[40]. Determining exactly how M1 polyubiquitination of RTA regulates its nuclear translocation will require further investigation. Moreover, considering there have been no other studies reporting that linear M1 ubiquitination regulates nuclear import, it will be interesting to see whether this PTM mediates the translocation of other nuclear proteins.

Given the current understanding that both LUBAC and OTULIN are localized in the cytoplasm, it is reasonable to assume that both the addition and the removal of M1 polyUb chains are carried out in the cytoplasm. M1 ubiquitination is a dynamic and reversible process. For

**Fig. 4 | OTULIN relocates RTA from the nucleus to the cytoplasm. a** M1 ubiquitinated RTA is enriched in nucleus. HeLa cells were co-transfected with RTA-Flag and Myc-OTULIN and subjected to fractionation into cytoplasmic and nuclear portions. RTA in the cytoplasm and nucleus was immunoprecipitated with anti-Flag magnetic beads and immunoblotted with an anti-linear Ub antibody. n = 3 independent experiments. **b, c** OTULIN facilitates relocation of RTA from the nucleus to the cytoplasm. HeLa cells were transfected with RTA together with OTULIN or OTULIN[C129A] (b) or iSLK/rKSHV.219 cells were transfected with OTULIN or OTULIN[C129A] and treated with Tet plus VPA (c). Cells were then harvested for fractionation into cytoplasmic and nuclear portions and immunoblotted with the indicated antibodies. *n* = 3 independent experiments. **d, e** OTULIN alters the intracellular localization of RTA. HEK293T cells were transfected with RTA-Flag alone (top panels), co-transfected with Myc-OTULIN (middle panel image) or with Myc-OTULIN[C129A] (lower panel image). After 36 hours, cells were subjected to immunofluorescence analysis. Subcellular localizations of RTA (in green) and of OTULIN (in red) are shown in (**d**) and quantification is shown in (**e**). Scale bar: 10 µm. All cell nuclei in the fields were stained with Hoechst (blue). Image is representative of 110 or more cells per condition in total *n* = 3 independent experiments. **f, g** RTA[2KR] is located in both the cytoplasm and the nucleus and OTULIN has no effect on its subcellular localization. HeLa cells were transfected with RTA[2KR] in the presence or absence of OTULIN or OTULIN[C129A] and harvested for fractionation into cytoplasmic and nuclear portions and subjected to immunoblotting with the indicated antibodies (**f**). HEK293T cells transfected with the indicated plasmids were subjected to immunofluorescence staining to assess the subcellular localization of RTA[2KR] (green). Cell nuclei in the fields were stained with Hoechst (blue) (**g**). Scale bar: 10 µm. Image is representative in total *n* = 3 independent experiments. Source data are provided as a Source Data file.

RTA, LUBAC ligates M1 polyUb chains onto it, whereas OTULIN removes the chains from it. Additionally, M1 polyubiquitinated RTA shuttles between the nucleus and cytoplasm, which also maintains dynamic equilibrium. The net status of RTA M1 ubiquitination is determined by these opposing activities. Understanding how these events are regulated and whether KSHV regulates these events during lytic reactivation requires further investigation.

Physiologically, OTULIN serves as a crucial molecular nexus for numerous cellular signaling pathways; medically, it is thought that OTULIN may contribute to angiogenesis[41,42]. KS tumors are characterized by proliferation of infected endothelial cells, pronounced neo-angiogenesis and vascularization. Considering the function of KSHV in modulating angiogenesis through autocrine and paracrine mechanisms by orchestrating a multifaceted interaction involving diverse pro-angiogenic factors[43,44], it may be fruitful to examine clinical KS specimens to investigate the possible pathogenic contributions of OTULIN and M1 ubiquitination to KSHV-mediated angiogenesis. Although it is still very early, our study raises the possibility that OTULIN or other M1 ubiquitination cellular machinery could be potential targets for developing therapies against diseases caused by KSHV infection.

Interestingly, in addition to our findings that M1 ubiquitination regulates protein subcellular localization, a recent study reported the role of M1 ubiquitination in regulating protein stability: LUBAC binds and stabilizes GPx4 by modulating its M1 ubiquitination under both normal and oxidative stress conditions[45]. We also noticed there was a slight increase in RTA levels alongside enhanced M1 polyubiquitination in some of our immunoblots for RTA, implying that M1 polyubiquitination might affect RTA protein stability. However, such results were not consistent across various experiments throughout our study. To be prudent we feel it is more appropriate not to conclude that M1 polyubiquitination of RTA might play a role in regulating its protein stability at this stage. Nonetheless, these studies highlight the complexity regarding the role of M1 and, in general, other linkage-specific ubiquitination events. Ubiquitination, as a critical post-translational modification, plays a critical role in regulating not only protein stability but also protein interactions and other biological functions. It is also crucial for a myriad of cellular processes including virus-host dynamics and immune responses. These diverse roles, to a large extent, depend on the type of ubiquitin linkage involved. While K48-linked ubiquitination is well-known for signaling proteasomal degradation, other linkages like K63 and M1 have been implicated in diverse cellular processes ranging from DNA repair to signaling pathways such as NF-κB activation, with less direct association with protein stability[23,46,47]. The interplay between ubiquitin linkages and protein stability, especially the potential impact of M1 ubiquitination on protein stability, merits further investigation.

In conclusion, our study established that linear polyubiquitination plays a crucial role in regulating KSHV lytic reactivation and infection. We revealed that LUBAC ubiquitinates RTA at K516 and K518, enabling its nuclear translocation and activation of the lytic gene cascade.

Conversely, OTULIN can reverse the modification from RTA and its nuclear translocation, therefore inhibiting KSHV lytic reactivation. We also reported here that under hypoxic condition, the M1 polyubiquitination machinery is involved in RTA regulation during KSHV lytic reactivation. Additionally, we demonstrated RTA orthologs from EBV and MHV68 shared the same regulatory mechanism by OTULIN. Our findings highlight the potential of intervening the M1 ubiquitination machinery in KSHV-associated diseases and the value of further in-depth research into the functions of M1 ubiquitination in viral infection.

## Methods

### Cell culture and chemical treatment

iSLK/rKSHV.219 cells were provided by Dr. Ke Lan (Wuhan University), Drs Chuan Lu and Qin Yan (Nanjing Medical University). BCBL-1 cells were provided by Dr. Ke Lan and Dr. Qiming Liang (Shanghai Jiao Tong University). HEK293T cells (CRL-3216), HeLa cells (CCL-2), and HUVECs (CRL-1730) were obtained from ATCC. iSLK/rKSHV.219 cells, HEK293T cells, HeLa cells, and HUVECs were maintained in DMEM medium (Gibco), BCBL-1 cells, in RPMI1640 medium (Gibco). All culture media were supplemented with fetal bovine serum (FBS, 10%, Gibco) and penicillin-streptomycin (1%, Gibco). To induce lytic replication, iSLK/rKSHV.219 cells were stimulated with tetracycline (1 µg/ml) and valproate (VPA, 1 mM, Sigma) for 48 hours. HUVEC cells post infection with rKSHV.219 were challenged with $CoCl_2$ (1 mM) for 48 hours. BCBL-1 cells were challenged with 12-O-tetra-decanoylphorbol-13- acetate (TPA, 20 ng/ml, Sigma) and NaB (0.3 mM, Sigma).

### Flow cytometry

To evaluate RFP[+] or GFP[+] cells via Flow cytometry, cells were washed with PBS, and resuspended with PBS, and analyzed on a FACSCalibur flow cytometer (BD). Data were processed with FlowJo software. Gating criteria were established by comparing to control cells.

### siRNA transfection

For transient silencing of OTULIN, siRNAs against OTULIN, or a control siRNA were transfected into iSLK/rKSHV.219 cells and BCBL-1 cells according to the manufacturer's instructions utilizing Lipofectamine RNAiMAX (Invitrogen). The sequences of the three siRNAs targeting OTULIN are: GACUGAAAUUUGAUGGGAAdTdT, CAAAUGAGGCGGAG GAAUAdTdT, and UAGCAAAGGCAGGGCGCAAdTdT.

### shRNAs targeting OTULIN

shRNAs targeting OTULIN were cloned into pLKO vector and used to make lentiviruses. HUVEC cells were inoculated with the recombinant lentiviruses and selected with puromycin to generate stable OTULIN knockdown cell lines. The three sequences of OTULIN shRNAs are: CCGGGGGCATCAGAACCGAGATTAAGCTCGAGCTTAATCTCGGTTCTG ATGCCTTTTTG for OTULIN shRNA 1, CCGGCCCTCATCTATGATGCAA

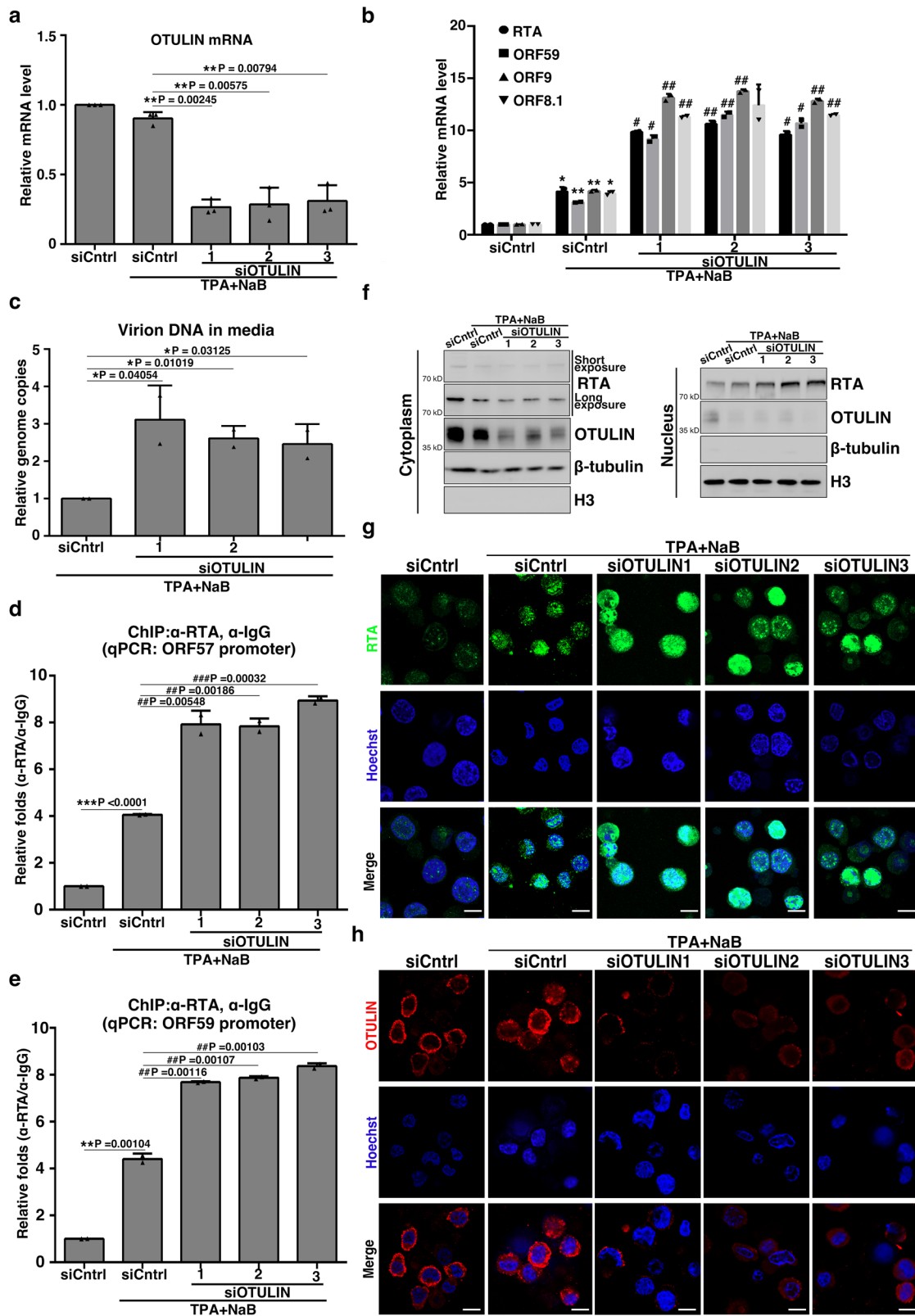

TATCTCGAGATAT-TGCATCATAGATGAGGGTTTTTG for OTULIN shRNA 2, and CCGGGCAATGGAA-ACTTGGACTGAACTCGAGTTCAG TCCAAGTTTCCATTGCTTTTTG for OTULIN shRNA 3.

## Infection of HEK293T cells and HUVEC cells with rKSHV
HEK293T cells or HUVEC cells were plated at $5 \times 10^5$ cells/ml in 6-well plates and inoculated with cell culture media obtained from iSLK/

rKSHV.219 cells stimulated with Tet and VPA at 37 °C for 4 hours. After inoculation, the media were changed to fresh DMEM, and cells were then maintained at 37 °C with 5% $CO_2$ for 48 hours.

## Quantitative detection of KSHV virions in cell culture media
Culture media was used to extract viral DNA using the AxyPrep™ Body Fluid Viral DNA/RNA Miniprep Kit (AXYGEN) following the

**Fig. 5 | Knockdown of OTULIN enhances KSHV lytic reactivation in BCBL-1 cells.**
**a** BCBL-1 cells were transfected with control siRNA (siCntrl) or three different siR-NAs against OTULIN (siOTULIN 1, 2 and 3). Knockdown efficiencies of OTULIN in BCBL-1 cells were measured by RT-qPCR. n = 3 biological replicates and 2 technical replicates, mean ± s.d., two-sided t test. **b** BCBL-1 cells from (**a**) were treated with TPA plus NaB for 24 h. The mRNA levels of *RTA*, *ORF59*, *ORF9*, and *ORF8.1* were determined using RT-qPCR. n = 2 biological replicates and 2 technical replicates, mean ± s.d., two-sided t test, * P = 0.02880, 0.00475, 0.00568, 0.015 (siCntrl vs. siCntrl+TPA+NaB) for RTA, ORF59, ORF9, and ORF8.1. #, P = (0.01953, 0.00380, 0.03093), (0.01167, 0.00975, 0.01240), (0.00778, 0.00557, 0.00764), (0.00265, 0.00877) for RTA, ORF59, ORF9, and ORF8.1. (**c**) As in (**b**), but the DNA in the culture media was extracted and qPCR was used to measure KSHV virions, n = 2 biological replicates and 2 technical replicates, mean ± s.d., two-sided t test.

**d**, **e** OTULIN knockdown enhances RTA binding to the *ORF57* and *ORF59* promoters. BCBL-1 cells were transfected with siCntrl or siOTULIN followed by treatment with TPA plus NaB. The cells were subjected to ChIP assay. The quantities of *ORF57* (**d**) or *ORF59* (**e**) promoter DNA in the precipitates were determined by qPCR. **d**, n = 2 biological replicates and 2 technical replicates, mean ± s.d., two-sided t test. **e**, n = 2 biological replicates and 2 technical replicates, mean ± s.d., two-sided t test. **f**–**h** OTULIN knockdown alters the intracellular localization of RTA. BCBL-1 cells were transfected with siCntrl or siOTULIN followed by treatment with TPA plus NaB. Cells were detected by immunoblotting (**f**) and immunofluorescence (**g**, **h**). n = 3 independent experiments. Subcellular localizations of RTA (in green) and of OTULIN (in red) are shown in (**g**) and (**h**). All cell nuclei in the fields were stained with Hoechst (blue). Scale bar: 10 μm. Image is representative in total n = 3 independent experiments. Source data are provided as a Source Data file.

manufacturer's guidance. For the purpose of normalization during the subsequent qPCR step, a fixed quantity of OTULIN plasmid was included in the media prior to DNA extraction. The extracted DNA was quantified for the KSHV LANA gene via qPCR using the following primer pairs: LANA-F, 5′-TCCAAAGTGTCAATGGAAGT, and LANA-R, 5′-GTAGATGGGTCGTGAGAACA. Relative levels were quantified with the ΔΔCT method (OTULIN plasmid was included as a control). Each sample was analyzed in triplicate.

### Subcellular fractionation
Nuclear fractions were prepared from HeLa, iSLK/rKSHV219 and BCBL-1 cells with the CelLytic NuCLEAR™ Extraction Kit (Sigma), in line with the manufacturer's instructions for cell collection, lysis, and sub-cellular fractionation. Both the nuclear and cytoplasmic fractions were then analyzed via Western blotting. Antibodies against actin (Huabio, R1207-1, 1:1000) and Histone H3 (Huabio, HA500298, 1:1000) were used to verify the efficiency of fractionation.

### Ubiquitination and deubiquitination assays
Following transfection with specified plasmid in HEK293T cells, cells were harvested and homogenized using a denaturing lysis buffer containing 6 M guanidine-HCl, 100 mM sodium phosphate buffer, pH 8.0, and 5 mM imidazole. After sonication, the lysates were centrifuged at 20,000 g for 10 min at 4 °C. The resulting supernatants were mixed with either M2 beads or protein G agarose beads immobilized with anti-RTA antibody for 4 h at 4 °C. The beads were then washed sequentially with denaturing lysis buffer at pH 8.0, denaturing lysis buffer at pH 6.0, and protein buffer (50 mM sodium phosphate buffer, pH 8.0, 100 mM KCl, 20% glycerol, 0.2% NP-40). Proteins bound to the beads were finally eluted using SDS-PAGE loading buffer at 95 °C and subjected to Western blot analysis.

### Quantitative reverse transcription-PCR (qRT-PCR)
Trizol reagent (Life Technologies) was used for Total RNA isolation following the manufacturer's instructions. RNA was then reverse-transcribed into cDNA with the RevertAid First Strand cDNA Synthesis Kit (Thermo) following the provided protocol. Genes interested were quantified with corresponding primers and SYBR® Premix Ex Taq™II (Tli RNaseH Plus) (TaKaRa) on a 7500 fast real-time PCR system (Applied Biosystems). The primer sequences used are listed as follows: forward primer, GTTAGCGGAATGTCTGTTTC, and reverse primer, ATGGTGTATGGCGATAGTGT for *vFlip*; forward primer, TATCCAGGA AGCGGTC- TCAT, and reverse primer, GGGTTAAAGGGGATGATGCT for *RTA*; forward primer, AACACCCTGATTACAAAAGC, and reverse primer, ATCAAAGTCCGAAACAGATG for *v-Cyclin*; forward primer, GACAGCTTCTGAGGAACCACCT, and reverse primer, TCCGTGTTGT ACTTGGAGAGCC for *OTULIN*; forward primer, GGGAAATCGTGC-GTGACAT, and reverse primer, GTCAGGCAGCTCGTAGCTCTT for *actin*; forward primer, TGGTCGGCGGTTCAGTCATCAA, and reverse primer, GCGGCCGCTAAG- AAAATCGA for *ORF8.1*; forward primer, TTAGAAGTGGAAGGTGTGCC, and reverse primer, TCCTGGAGTCCGG

TATAGAATC for *ORF59*; forward primer, TAGGCGCTTCGTGCTGG, and reverse primer, CCGGATTGCTGCACTCGTA for *ORF9*. Relative expression levels of the above genes were calculated using the ΔΔC$_T$ method by normalization to *actin*. Each sample was conducted in triplicates.

### Luciferase reporter assay
The Dual Luciferase reporter assay system (Promega) were used to quantify RTA transactivation activity. Briefly, cells were co-transfected with luciferase reporter plasmids and relevant plasmids and cell lysates were generated by following the manufacturer's protocol. Luciferase activities were quantified on a GloMax-Multi Microplate Multimode Reader (Promega). Data were presented as the ratio of firefly to Renilla luciferase activities and the results were derived from experiments conducted in duplicates.

### ChIP assays
Cells were incubated with 1% formaldehyde (Thermo Scientific) for 10 min at room temperature for cross-linking. The cross-linking reaction was terminated by adding glycine to a final concentration of 125 mM and incubating for an additional 5 minute at room temperature. After two washes with pre-cold PBS, cells were homogenized in FastChIP buffer (50 mM Tris-HCl, pH 7.5, 150 mM NaCl, 0.5 mM DTT, 5 mM EDTA, 0.5% IGEPAL CA-630, 1.0% Triton X-100) supplemented with protease inhibitors. Low-speed centrifugation was performed to pellet nuclei, which were then resuspended in SDS lysis buffer (50 mM Tris-HCl pH 8.1, 10 mM EDTA, 1% SDS), and maintained on ice for 15 min. Following sonication and centrifugation at 20,000 g and 4 °C for 30 minutes, the supernatants were collected, diluted with 9 volumes of ChIP dilution buffer (16.7 mM Tris-HCl pH 8.1, 167 mM NaCl, 1.2 mM EDTA, 0.01% SDS, 1.1% Triton X-100), and used for subsequent immuno-precipitation using the following antibodies: mouse anti-RTA antibody (8C12, Argene, 1:1000), anti-Flag antibody (Huabio, 0912-1, 1:1000), or rabbit IgG. After incubation at 4 °C for 12 hours, protein A/G beads (Thermo Scientific) was added and incubated for another 3 hours. Following sequential washes with low-salt buffer (20 mM Tris-HCl pH 8.1, 150 mM NaCl, 2 mM EDTA, 0.1% SDS, 1% Triton X-100), high-salt buffer (20 mM Tris-HCl pH 8.1, 0.5 M NaCl, 2 mM EDTA, 0.1% SDS, 1% Triton X-100), LiCl buffer (10 mM Tris-HCl pH 8.1, 0.25 M LiCl, 1 mM EDTA, 1% deoxycholic acid, 1% IGEPAL CA-630), and twice with TE buffer (10 mM Tris-HCl pH 8.1, 1 mM EDTA), the precipitates were resuspended in TE buffer supplemented with 0.2 M NaCl, 1% SDS, and protease K (Roche) for digestion for 12 hours at 65 ˚C. DNA in the resulting solution was purified using a DNA gel extract kit (AXYGEN) used as the template for qPCR. Primers used were listed as follows: *ori-Lyt* left (forward primer, 5′-CCCTCCTTTGTTTTCCGGAAG-3′, reverse primer, 5′-CTCATCGGGCCCTATTATAAAG-3′), *RTA* promoter (forward primer, 5′-GAACTACTCGAGCTGTGCCCTCCAGCTCTCAC-3′, reverse primer, 5′-GGACGTAAGCTTACAGTATTCTCACAACAGA

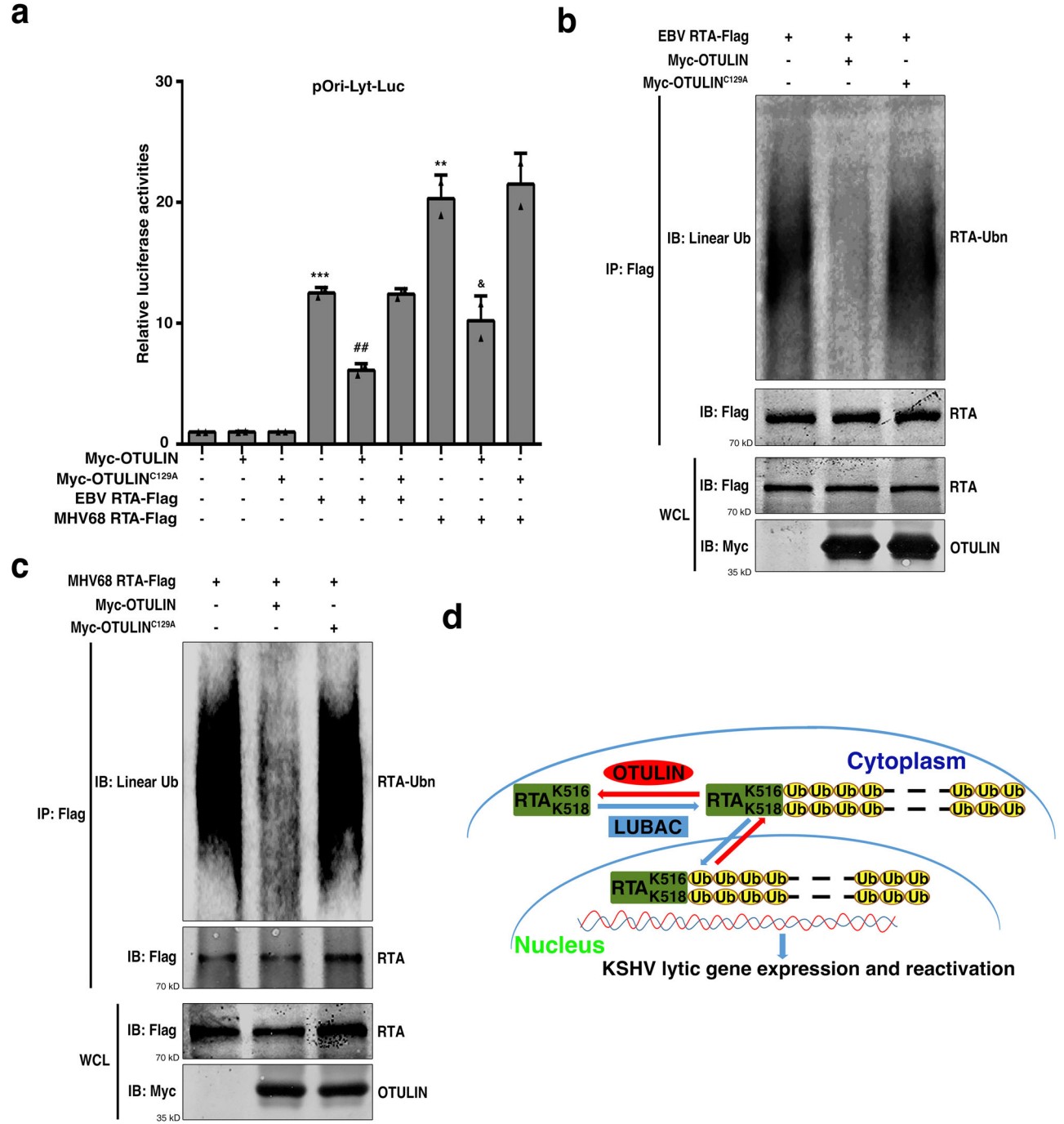

**Fig. 6 | OTULIN inhibits the transcriptional activities and M1 ubiquitination of RTAs encoded by EBV and MHV68. a** OTULIN inhibits the transcriptional activities of EBV RTA and MHV68 RTA. HEK293T cells were transfected with the firefly luciferase reporter construct pGL3-Ori-Lyt-Luc, a Renilla luciferase control plasmid, RTA by EBV or MHV68, together with OTULIN or OTULIN^C129A. After 36 h cells were harvested and subjected to measurement of luciferase activities, *n* = 2 biological replicates and 2 technical replicates, mean ± s.d., two-sided t test, * P = 0.00041 (Cntrl vs. EBV), * P = 0.00252 (Cntrl vs. MHV68) #, P = 0.00313 (EBV vs. EBV + OTULIN), &, P = 0.01841 (MHV68 vs. MHV68 + OTULIN). **b**, **c** OTULIN diminishes M1 ubiquitination of EBV RTA and MHV68 RTA. Flag tagged EBV RTA or MHV68 RTA,

together with Myc-OTULIN or OTULIN^C129A was expressed in HEK293T cells and immunoprecipitated with anti-Flag magnetic beads. M1 ubiquitination of EBV RTA and MHV68 RTA were determined by immunoblotting with a specific anti-linear Ub antibody. *n* = 3 independent experiments. **d** A model for the M1-polyubiquitination-mediated regulation of RTA's role in KSHV lytic reactivation. LUBAC specifically ubiquitinates RTA on K516 and K518, which enables RTA translocation into nucleus where it can activate the expression of a lytic gene cascade that initiates KSHV lytic reactivation. Opposing this process, OTULIN can cleave the linear ubiquitination modification from RTA and thereby inhibit its nuclear translocation, thus inhibiting KSHV lytic reactivation. Source data are provided as a Source Data file.

C-3'), *PAN* promoter (forward primer, 5'-CTACTAGCTAGCGTTTATT AATGTTCATCCGTATTGTG-3', reverse primer, 5'-CTAGCCAAGCTT CTGGGCAGTCCCAGTGCTAAAC-3'), *ORF57* promoter (forward primer, 5'-CTACTAGCTAGCCAAGACCATTAGCTATCTGCC-3', reverse primer, 5'-CGACCCAAGCTTGGGCTATTTTGGGAACCTGG-3').

## Immunofluorescence

HEK293T and BCBL-1 cells were plated directly onto coverslip in cell culture plates, incubated overnight, and transfected the following day. At 24 hours post-transfection or post-induction, the cells were fixed, subjected to blocking, and then incubated with specified primary

antibody for RTA or OTULIN. After 1 hour of incubation, the cells were washed three times and before incubation with specified secondary antibodies (either fluorescein isothiocyanate- or Texas Red-conjugated). After 1 hour of incubation, the cells were washed and stained with Hoechst. Finally, the cells were washed and imaged using fluorescence microscopes.

## Immunoblotting

Cells were lysed using the lysis buffer (150 mM NaCl, 20 mM Tris-HCl pH 7.4, 0.5 mM EDTA, 1% NP-40, 1 mM PMSF, and protease inhibitor cocktail (Roche)) and centrifuged at 20,000 x $g$ for 15 minutes. Cleared supernatant were subjected to protein concentration measurement. Equal contents of lysates were subjected to SDS-PAGE separation and transferred onto PVDF membranes. The membranes were incubated with 5% BSA for 1 hour at room temperature for blocking. To detect proteins interested, the membranes were incubated sequentially with corresponding primary antibodies and HRP-conjugated secondary antibodies. Bands were visualized using SuperSignal West Dura Extended kits (Thermo Scientific). The antibodies used were as follows: RTA (8C12, Argene, 1:1000), Myc (Huabio, 0912-2, 1:1000), Flag (Huabio, 0912-1, AB_3068715), Histone H3 (Huabio, HA500298, AB_3071393), β-actin (Huabio, R1207-1, AB_3073201), Linear Ubiquitin (Millipore, MABS199, 1:1000), OTULIN (Abcam, ab151117, 1:1000) and GAPDH (Cell Signaling Technology, 2118, 1:10000).

## Co-immunoprecipitation (co-IP)

To detect the interaction between RTA and OTULIN, HEK293T cells were transfected with RTA and OTULIN plasmids and cultured for 48 hours. The cells were harvested and lysed in buffer A (50 mM Tris-Cl, pH 7.4, 150 mM NaCl, 0.5% NP-40, 10% Glycerol, 1 mM PMSF, and 1 mM DTT). The cleared lysate was incubated with M2 beads (Sigma) to capture protein complex, which was then detected by Western blotting using specific antibodies.

To detect the interaction between endogenous OTULIN and RTA, iSLK/rKSHV.219 cells were stimulated with tetracycline (1 μg/ml) and VPA (1 mM, Sigma) to induce RTA expression. After 48 hours, the cells were harvested and lysed in buffer A, and the cleared lysate was incubated with anti-RTA antibody. Protein G-coupled agarose beads (Sigma) were then added to precipitate the complexes, which were then subjected to Western blotting to detect RTA and OTULIN.

## Statistical analysis

Bar graphs were displayed with mean values and standard deviations of a minimum of three biological replicates. p-values were quantified in GraphPad Prism with unpaired two-tailed t-tests. *, $P < 0.05$, **, $P < 0.01$, ***, $P < 0.001$.

## Reporting summary

Further information on research design is available in the Nature Portfolio Reporting Summary linked to this article.

# Data availability

The datasets are available within the Article, Supplementary Information, or Source Data file. Source data are provided as a Source Data file. Source data are provided with this paper.

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

## Acknowledgements

This work was partially funded by the National Natural Science Foundation of China (31970158 and 31571445 to Z.X.). We thank Drs. Ke Lan, Chuan Lu, Qin Yan, and Qiming Liang for the iSLK/rKSHV.219 and BCBL-1 cell lines, and the RTA antibodies; Drs. Musheng Zeng and Hongyu Deng for the EB RTA and MHV68 RTA constructs; and Dr. Fanxiu Zhu for his valuable insights.

## Author contributions

Y.L., W.L., and Z.X. conceived and designed the study, analyzed all the data, and co-wrote the manuscript. Y.L. and W.L. performed all the experiments. L.D., P.T., and Z.D. helped with cell culture and some reporter assays.

## Competing interests

The authors declare no competing interests.
