## [Peer Review File · Nature Communications]

REVIEWER COMMENTS

Reviewer #1 (Remarks to the Author):

This study by Luan et al. explore a specific less known posttranslation modification, called linear Met1-linked polyubiquitination (M1 polyUb) and the role of this modification in Kaposi's sarcoma-associated herpesvirus (KSHV) life cycle. Since the exact role of this modification is still largely unknown in the KSHV field, therefore the topic of this study is of high importance. The study goes on to describe a new regulatory modification of a key KSHV factor called RTA. RTA is a central viral protein and necessary to promote lytic cycle of KSHV. The authors present compelling evidence for the role of a host deubiquitinase (DUB) enzyme called OTULIN as well as for the opposing proteins, forming the E3 ligase linear Ub chain assembly complex (LUBAC), namely HOP/HOIL-1L factors. The specific LUBAC components were described to ubiquitinate RTA and the specific residues were identified. It was demonstrated that the addition and removal of this modification from RTA regulates the subcellular location of RTA. Furthermore, OTULIN was described to remove these marks from RTA and thereby regulate the transcription activity of RTA using reporter assays. The authors go on to show that the overexpression OTULIN inhibits KSHV lytic reactivation while depletion induces it. Overall, the data presented is well organized, and the findings are generally of importance. In the following, I provide specific points, which should be addressed to strengthen the model described in the manuscript:

Major points:

1. Authors do not provide evidence that the endogenous RTA location would be affected by the loss of OTULIN, which would majorly strengthen the manuscript and validate the working model. To this end, authors should provide evidence that the endogenous viral RTA subcellular localization changes during lytic reactivation in a biologically relevant cell (such as BCBL1) upon depletion of the OTULIN, to provide additional evidence for their proposed model. Advanced imaging approaches and biochemical fractionation would be required to show RTA location changes upon siRNA or shRNA depletion of OTULIN, which could be added to the experiment in Fig 5.
2. Similarly, to the above comment, the authors should also test if OTULIN depletion affects endogenous RTA binding to the actual KSHV genome by ChIP-qPCR, which would be important as it would majorly strengthen the current conclusions based on the data using plasmids.
3. Authors should also include an immunoblot analysis on Fig 5 to show how efficient is the OTULIN depletion by siRNA knockdown in BCBL1 cells.
4. The authors use the statement "significant" during the description of their findings at multiple location (such as row #130, 139, 141 etc.). The statement should be supported by statistical

analysis and revised as needed. The manuscript does not contain statistical analysis, which is required for interpretation for some of the datasets, thus should be included in the analysis as appropriate, and described in the manuscript.

Minor points:

1. Missing axis labels on graphs showing flow cytometric evaluations (such as Fig 1 B, E. and supFig5B)
2. Unclear value / Y axis on the graph showing RT-qPCR results (Fig 2B). Y axis indicates “Relative viral gene expression”, however siCntrl is not indicated to be 1. Authors should clarify the calculation and check axis.
3. Based on the rest of the table design, 4th column (from the last) might be mislabelled on Sup fig 3A, as it indicates 2 RTA constructs (instead of 1). The large table under the graph should be checked, and if needed, corrected.
4. Fig 4A minor label correction needed to indicate “linear Ub” on the top part of the figure, instead of the “linar Ub” text , as shown currently.
5. Sup Fig 5D, the RFP+ cell number is difficult to observe on the images, it should be quantified similarly to the Sup Fig 5B (showing GFP+ rate).

Reviewer #2 (Remarks to the Author):

Met1-linked polyubiquitin (Met1 polyUb) chains are formed when the amino terminus of one ubiquitin is conjugated to the carboxyl terminus of another ubiquitin. HOIP E3 ligase is the only known ligase capable of generating M1 polyUb chains. However, a cellular deubiquitination enzyme OTULIN specifically blocks the formation of M1 polyUb chains and thus prevents E3 ligase activity. Authors In this study discovered M1 ubiquitination at RTA K516 and K518 residues during KSHV infection and this ubiquitination on RTA is essential for KSHV replication and virus production. This was concluded by blockade of M1 ubiquitination on RTA by overexpression OTULIN or knockdown of endogenous OTULIN expression. The M1 ubiquitination of RTA being necessary for KSHV replication and virus production was also verified by E3 ligase HOIP. More interestingly, although RTA itself is an E3 ligase which mediates protein degradation of host targets such as IRF7 and its own stability, the authors showed that deubiquitinated RTA at K516 and K518 was translocated to the cytoplasm from the nucleus in a dose-dependent manner of OTULIN, thus, reducing the

expression of viral lytic genes. Overall, this study provides novel observations on KSHV RTA regulation and was well designed experimentally. The results support authors' conclusion. Other specific comments are in the followings:

1. Fig. 2C, it would be better to show a lower exposure of Western blot for increased OTULIN protein dose in correlation with decreased luciferase activity.
2. Fig. 3. It is obvious that RTA interacts linear Ub in the cells and its ubiquitinations happen at K516 and K518. Although RTA level in Fig. 3C was increased in the presence of wt Ub in the whole cell lysis, Consistently, the data in Fig. 3F with RTA2KR appear loss of M1 ubiquitination on RTA2KR prevents ubiquitination-mediated RTA stability. However, a lower exposure for IB RTA in the IP blot in Fig. 3F may be indicative. It would be better also to reduce the exposure for linear Ub blots of Fig. 3A, 3B, 3C, 3D, and 3F.
3. Fig. 4D-E and F-G. Which cell type was used in the study, HeLa or HEK293? Text did not mention HEK293T, but only HeLa cells. Fig. 4F indicates HeLa, but Fig. 4G legend indicates HEK293.
4. Fig. 6B. Please reduce the exposure for linear Ub blot
5. Lines 47-48: citations were wrong. Ref. #6 was about TPA and NaB and Ref. 7 about hypoxia induced KSHV lytic infection. None of these two references had any thing to do with valproate or valproic acid (VA) induction of KSHV lytic infection. The initial publications on valproate induction of KSHV lytic infection but the inducer itself other than TPA and NaB which promotes only a minimal cell death, were Shaw RN, et al. AIDS 14:899-902, 2000; Klass CM, et al. Blood 105: 4028-34, 2005; Majerciak V., et al. JVI 81: 1062-1071, 2007.
6. Lines 289-291. "If M1 polyubiquitinated RTA can translocate back into the cytoplasm, then OTULIN can cleave its M1 polyUb chains, preventing further nuclear translocation....". As both M1 polyubiquitination and deubiquitination happen in the cytoplasm, these are two competitive events at the same time and should happen when RTA is translated in the cytoplasm. Why RTA deubiquitination should wait for the RTA being translocated back to the cytoplasm for OTULIN function. It is obvious that M1 ubiquitination of RTA is required for nuclear location, but not another way around. Thus, this paragraph should be rewrite and discussed further, including RTA ubiquitination and protein degradation in the cytoplasm as shown in Fig. 4.
7. Abstract. "OTULIN removed M1 chains from RTA, enabling RTA's nucleus-to-cytoplasm relocation" should be changed to "OTULIN removed M1 chains from cytoplasmic RTA, preventing its nuclear import".

REVIEWER COMMENTS

Reviewer #1 (Remarks to the Author):

This study by Luan et al. explore a specific less known posttranslation modification, called linear Met1-linked polyubiquitination (M1 polyUb) and the role of this modification in Kaposi's sarcoma-associated herpesvirus (KSHV) life cycle. Since the exact role of this modification is still largely unknown in the KSHV field, therefore the topic of this study is of high importance. The study goes on to describe a new regulatory modification of a key KSHV factor called RTA. RTA is a central viral protein and necessary to promote lytic cycle of KSHV. The authors present compelling evidence for the role of a host deubiquitinase (DUB) enzyme called OTULIN as well as for the opposing proteins, forming the E3 ligase linear Ub chain assembly complex (LUBAC), namely HOP/HOIL-1L factors. The specific LUBAC components were described to ubiquitinate RTA and the specific residues were identified. It was demonstrated that the addition and removal of this modification from RTA regulates the subcellular location of RTA. Furthermore, OTULIN was described to remove these marks from RTA and thereby regulate the transcription activity of RTA using reporter assays. The authors go on to show that the overexpression OTULIN inhibits KSHV lytic reactivation while depletion induces it. Overall, the data presented is well organized, and the findings are generally of importance. In the following, I provide specific points, which should be addressed to strengthen the model described in the manuscript:

Response: Thank you very much for your constructive and insightful feedback. We have carefully reviewed and summarized your constructive comments, and carefully revised the figures and manuscript, implementing related experiments, enriching part of the content, citing the latest studies, and integrating part of the content to improve logic and make the manuscript more coherent. We believe that the revised version has been greatly improved in the quality and readability. We look forward to your valuable suggestions you may have on this revised version.

Major points:

1. Authors do not provide evidence that the endogenous RTA location would be affected by the loss of OTULIN, which would majorly strengthen the manuscript and validate the working model. To this end, authors should provide evidence that the endogenous viral RTA subcellular localization changes during lytic reactivation in a biologically relevant cell (such as BCBL1) upon depletion of the OTULIN, to provide additional evidence for their proposed model. Advanced imaging approaches and biochemical fractionation would be required to show RTA location changes upon siRNA or shRNA depletion of OTULIN, which could be added to the experiment in Fig 5.

Response: Thank you very much for your invaluable suggestion. Following your suggestion to provide biologically more relevant data for our proposed model, we

conducted additional experiments to evaluate endogenous viral RTA localization changes during lytic reactivation in BCBL-1 cells following OTULIN depletion via siRNA. Prior to lytic induction, RTA were marginally present in both cytosolic and nuclear fractions, with longer exposure time to show cytosolic signals. Upon induction with TPA and NaB, RTA levels were increased in both cytosolic and nuclear fractions, which were further elevated in the nuclear fraction upon OTULIN knockdown (Figure 5F, biochemical fractionation assay). Concurrently, through immunofluorescence staining, we assessed RTA subcellular distribution in BCBL-1 cells. Given the cells' spherical morphology and the limited cytosolic space, it was a little bit challenge to discern cytosolic RTA signals. Nevertheless, as illustrated in Figure 5G, RTA was detected in the nucleus of control cells, which was further intensified in OTULIN-depleted cells upon TPA and NaB induction (Figure 5G). These results strongly support our working model, demonstrating that loss of OTULIN promotes increased nuclear translocation of RTA in KSHV-infected cells.

2. Similarly, to the above comment, the authors should also test if OTULIN depletion affects endogenous RTA binding to the actual KSHV genome by ChIP-qPCR, which would be important as it would majorly strengthen the current conclusions based on the data using plasmids.

Response: Thank you very much for your insightful suggestion. As suggested, we analyzed whether OTULIN depletion would affect endogenous RTA binding to the actual KSHV genome by ChIP-qPCR analyses in BCBL-1 cells. We observed that OTULIN knockdown enhanced the interaction of endogenous RTA with the promoters of ORF57 (Figure 5D) and ORF59 (Figure 5E) in reactivated lytic BCBL-1 cells, suggesting that OTULIN depletion strengthened RTA binding to KSHV genome.

3. Authors should also include an immunoblot analysis on Fig 5 to show how efficient is the OTULIN depletion by siRNA knockdown in BCBL1 cells.

Response: We apologize for the oversight of not including an immunoblot analysis in Figure 5 in our previous version. The knockdown efficiency of OTULIN have now been verified by immunoblot analysis and added in Figure 5F.

We have presented the revised Figure 5 below for your convenient reference.

Figure 5. Knockdown of OTULIN enhances KSHV lytic reactivation in BCBL-1 cells. (A) Knockdown efficiencies of OTULIN in BCBL-1 cells were measured by RT-qPCR. (B) BCBL-1 cells from (A) were treated with TPA plus NaB for 24 hours. The mRNA levels of KSHV lytic genes RTA, ORF59, ORF9, and ORF8.1 were determined using RT-qPCR. (C) As in (B), but the DNA in the culture media was extracted and qPCR was used to measure KSHV virions. (D, E) OTULIN knockdown enhances RTA binding to KSHV ORF57 and ORF59 promoters. (F-G) OTULIN knockdown enhanced RTA nuclear localization determined by biochemical subcellular fractionation (F) and immunofluorescence staining (G). (H) Immunofluorescence staining of OTULIN.

4. The authors use the statement “significant” during the description of their findings at multiple location (such as row #130, 139, 141 etc.). The statement should be supported by statistical analysis and revised as needed. The manuscript does not contain statistical analysis, which is required for interpretation for some of the datasets, thus should be included in the analysis as appropriate, and described in the manuscript.

Response: Thank you very much for pointing out our ignorance. We agree with you that statements about significance should be supported by statistical analysis. We have now performed statistical analyses on all relevant datasets and updated the figures and main text accordingly in this revised manuscript.

Minor points:

1. Missing axis labels on graphs showing flow cytometric evaluations (such as Fig 1 B, E. and supFig5B)

Response: Thank you very much for your kind suggestions. We have now incorporated the axis labels on graphs to show flow cytometric evaluations in Figures 1B, 1E, and supplemental Figures 5B & 5E, as well as others in the revised version.

2. Unclear value / Y axis on the graph showing RT-qPCR results (Fig 2B). Y axis indicates “Relative viral gene expression”, however siCntrl is not indicated to be 1. Authors should clarify the calculation and check axis.

Response: We apologize for the oversight regarding Y-axis values related to the siCntrl group. The relative viral gene expression in the siCntrl group without Tet+VPA treatment should be set to 1. We have carefully checked our original data and updated Figure 2B in the revised version. The corrected Figure 2B is also provided below for your quick reference.

B

3. Based on the rest of the table design, 4th column (from the last) might be mislabelled on Sup fig 3A, as it indicates 2 RTA constructs (instead of 1). The large table under the graph should be checked, and if needed, corrected.

Response: Thank you very much for your careful inspection and pointing out the mislabeling in Supplementary Figure 3A, which was indeed incorrectly labeled. It should be RTA^{K530R} instead of RTA^{2KR} constructs. We have corrected the error in the revised version of this figure. We have also carefully checked the table under the graph to ensure its accuracy.

A

4. Fig 4A minor label correction needed to indicate “linear Ub” on the top part of the figure, instead of the “linar Ub” text, as shown currently.

Response: We apologize for the typographical error in Figure 4A. We have corrected it in the revised version of this figure. We have also carefully examined the labeling in all other figures to ensure accuracy.

5. Sup Fig 5D, the RFP+ cell number is difficult to observe on the images, it should be quantified similarly to the Sup Fig 5B (showing GFP+ rate).

Response: We really appreciate your insightful suggestion. We agree with you that it should be quantified regarding RFP-positive cells in Supplementary Figure 5D. We had actually conducted FACS analysis on the same samples shown in Supplementary Figure 5D but did not include the results in the previous version. We have now included the results as Supplementary Figure 5E in the revised Supplementary Figure 5, which is also shown below for your convenient reference. We found that HUVEC cells with OTULIN depletion displayed notable increases in the proportion of RFP+ cells than those with control shRNA (Figure S5, D and E).

Reviewer #2 (Remarks to the Author):

Met1-linked polyubiquitin (Met1 polyUb) chains are formed when the amino terminus of one ubiquitin is conjugated to the carboxyl terminus of another ubiquitin. HOIP E3 ligase is the only known ligase capable of generating M1 polyUb chains. However, a cellular deubiquitination enzyme OTULIN specifically blocks the formation of M1 polyUb chains and thus prevents E3 ligase activity. Authors In this study discovered M1 ubiquitination at RTA K516 and K518 residues during KSHV infection and this ubiquitination on RTA is essential for KSHV replication and virus production. This was concluded by blockade of M1 ubiquitination on RTA by overexpression OTULIN or knockdown of endogenous OTULIN expression. The M1 ubiquitination of RTA being necessary for KSHV replication and virus production was also verified by E3 ligase HOIP. More interestingly, although RTA itself is an E3 ligase which mediates protein degradation of host targets such as IRF7 and its own stability, the authors showed that deubiquitinated RTA at K516 and K518 was translocated to the cytoplasm from the nucleus in a dose-dependent manner of OTULIN, thus, reducing the expression of viral lytic genes. Overall, this study provides novel observations on KSHV RTA regulation and was well designed experimentally. The results support authors' conclusion. Other specific comments are in the followings:

Response: Thank you very much for your constructive suggestions. We have carefully revised the figures and manuscript, replacing part of the figures, enriching part of the content, citing the latest studies, and integrating part of the content to improve logic and make the manuscript more coherent. We believe that the current version better supports our conclusions and kindly submit it for your review.

1. Fig. 2C, it would be better to show a lower exposure of Western blot for increased OTULIN protein dose in correlation with decreased luciferase activity.

Response: Thank you very much for your valuable suggestion. Following your advice, we have updated Figure 2C in the revised version with a lower exposure of Western blot, which better demonstrates the inverse correlation between increased OTULIN protein dose and decreased luciferase activity. We also provided Figure 2C below for your reference.

2. Fig. 3. It is obvious that RTA interacts linear Ub in the cells and its ubiquitarians happen at K516 and K518. Although RTA level in Fig. 3C was increased in the presence of wt Ub in the whole cell lysis, Consistently, the data in Fig. 3F with RTA2KR appear loss of M1 ubiquitination on RTA2KR prevents ubiquitination-mediated RTA stability. However, a lower exposure for IB RTA in the IP blot in Fig. 3F may be indicative. It would be better also to reduce the exposure for linear Ub blots of Fig. 3A, 3B, 3C, 3D, and 3F.

Response: Thank you very much for your valuable suggestion. We have replaced the linear Ub blots at lower exposure levels in Figure 3A, 3B, 3C, 3D, and 3F. Meanwhile, a lower exposure for IB RTA in the IP blot in Figure 3F is also provided to reflect the influence of RTA^{2KR} on ubiquitination-mediated RTA stability.

We agree with you that in some of the WB analyses related to RTA (for example, in Figure 3A, 3B and 3C), there was a slight increase in RTA levels alongside enhanced M1 polyubiquitination, implying that M1 polyubiquitination might affect RTA protein stability. However, such results were not consistent across various experiments throughout our study. To be prudent we feel it is more appropriate not to conclude that M1 polyubiquitination of RTA might play a role in regulating its protein stability at this stage.

Notably, a recent publication reported the role of M1 ubiquitination in protein stability: LUBAC binds and stabilizes GPx4 by modulating its M1 ubiquitination under both normal and oxidative stress conditions (Proc Natl Acad Sci U S A. 2022 Nov;119(44):e2214227119). This work illustrates complex regulatory roles of M1 ubiquitination in cellular homeostasis under various pathophysiological conditions.

Nonetheless, we recognize the importance of this preliminary observation and will pay close attention to the potential role of M1 ubiquitination in RTA protein stability in our follow-up studies.

3. Fig. 4D-E and F-G. Which cell type was used in the study, HeLa or HEK293? Text did not mention HEK293T, but only HeLa cells. Fig. 4F indicates HeLa, but Fig. 4G legend indicates HEK293.

Response: Thank you very much for your valuable comments and we are really sorry for missing the details about cell types used in these figures and the corresponding text. In Figure 4D-E, HEK293T cells were used for immunofluorescent assay. And the subcellular localizations of RTA mutants RTA^{2KR} and RTA4^{KR} were determined in both HeLa (biochemical fractionation, Figure 4F and Supplementary Figure 4D) and HEK293T cells (immunofluorescence staining, Figure 4G and Supplementary Figure 4E). We have clarified the information in the revised figure legends and text.

4. Fig. 6B. Please reduce the exposure for linear Ub blot.

Response: Thank you very much for your valuable suggestion. Following your suggestion, we have replaced it with a shorter exposure image in Figure 6B.

5. Lines 47-48: citations were wrong. Ref. #6 was about TPA and NaB and Ref. 7 about hypoxia induced KSHV lytic infection. None of these two references had anything to do with valproate or valproic acid (VA) induction of KSHV lytic infection. The initial publications on valproate induction of KSHV lytic infection but the inducer itself other than TPA and NaB which promotes only a minimal cell death, were Shaw RN, et al. AIDS 14:899-902, 2000; Klass CM, et al. Blood 105: 4028-34, 2005; Majerciak V., et al. JVI 81: 1062-1071, 2007.

Response: Thank you very much for pointing out our inappropriate citations. We are really sorry for this mistake. We have corrected the citations by adding the suggested references in the revised version.

The corresponding revised content is as follows:

“KSHV mostly persists in the latent state during which it has a restricted latent gene expression program but can be reactivated and transitioned to the lytic state when triggered by stress conditions such as hypoxia or HIV coinfection, or stimulated by other chemical agents such as 12-O-tetradecanoylphorbol-13-acetate (TPA), sodium butyrate (NaB), and valproate (VPA) ^{6,7,8}.

6. Shaw RN, Arbiser JL, Offermann MK. Valproic acid induces human herpesvirus 8 lytic gene expression in BCBL-1 cells. *Aids* 14, 899-902 (2000).

7. Klass CM, Krug LT, Pozharskaya VP, Offermann MK. The targeting of primary effusion lymphoma cells for apoptosis by inducing lytic replication of human herpesvirus 8 while blocking virus production. *Blood* 105, 4028-4034 (2005).

8. Majerciak V, Pripuzova N, McCoy JP, Gao SJ, Zheng ZM. Targeted disruption of Kaposi's sarcoma-associated herpesvirus ORF57 in the viral genome is detrimental for the expression of ORF59, K8alpha, and K8.1 and the production of infectious virus. *Journal of virology* 81, 1062-1071 (2007).”

6. Lines 289-291. “If M1 polyubiquitinated RTA can translocate back into the cytoplasm, then OTULIN can cleave its M1 polyUb chains, preventing further nuclear translocation....”. As both M1 polyubiquitination and deubiquitination happen in the cytoplasm, these are two competitive events at the same time and should happen when RTA is translated in the cytoplasm. Why RTA deubiquitination should wait for the RTA being translocated back to the cytoplasm for OTULIN function. It is obvious that M1 ubiquitination of RTA is required for nuclear location, but not another way around. Thus, this paragraph should be rewrite and discussed further, including RTA ubiquitination and protein degradation in the cytoplasm as shown in Fig. 4.

Response: Thank you very much for highlighting areas that required clarification and for your valuable suggestions. We agree with you that the previous discussion is not appropriate and may have oversimplified the dynamic nature of RTA M1 ubiquitination. We have rewritten it in the revised version, which is also copied below.

For the dynamic regulation of RTA M1 polyubiquitination, our revised version is as follows:

“Given the current understanding that both LUBAC and OTULIN are localized in the cytoplasm, it is reasonable to assume that both the addition and the removal of M1 polyUb chains are carried out in the cytoplasm. M1 ubiquitination is a dynamic and reversible process. For RTA, LUBAC ligates M1 polyUb chains onto it, whereas OTULIN removes the chains from it, which represent dynamic and competing actions. Additionally, M1 polyubiquitinated RTA shuttles between the nucleus and cytoplasm, which also maintains dynamic equilibrium. The net status of RTA M1 ubiquitination is determined by these opposing activities. Understanding how these events are regulated and whether KSHV regulates these events during lytic reactivation requires further investigation.”

Regarding the putative role of M1 polyubiquitination in regulating RTA protein stability, as described in our response to your Point 2, we feel it would be prudent at this stage not to conclude that M1 polyubiquitination of RTA might play a role in regulating its protein stability.

Considering the potential of M1 ubiquitination in regulating protein stability, we added a whole paragraph discussing ubiquitination in the revised Discussion section, which is copied below:

“Interestingly, in addition to our findings that M1 ubiquitination regulates protein subcellular localization, a recent study reported the role of M1 ubiquitination in regulating protein stability: LUBAC binds and stabilizes GPx4 by modulating its M1 ubiquitination under both normal and oxidative stress conditions ⁴⁵. We also noticed there was a slight increase in RTA levels alongside enhanced M1 polyubiquitination in some of our immunoblots for RTA, implying that M1 polyubiquitination might affect RTA protein stability. However, such results were not consistent across various experiments throughout our study. To be prudent we feel it is more appropriate not to conclude that M1 polyubiquitination of RTA might play a role in regulating its protein stability at this stage. Nonetheless, these studies highlight the complexity regarding the role of M1, and in general other linkage ubiquitination events. Ubiquitination, as a critical posttranslational modification, plays a critical role in regulating not only protein stability but also its interactions and other biological functionality. It is also crucial for a myriad of cellular processes including virus-host dynamics and immune responses. These diverse roles, to a large extent, depend on the type of ubiquitin linkage involved. While K48-linked ubiquitination is well-known for signaling proteasomal degradation, other linkages like K63 and M1 have been implicated in diverse cellular processes ranging from DNA repair to signaling pathways such as NF- κ B activation, with less direct association with protein stability ^{46, 47, 48}. The interplay between ubiquitin linkages and protein stability, especially the potential impact of M1 ubiquitination on protein stability, merits further investigation.”

7. Abstract. “OTULIN removed M1 chains from RTA, enabling RTA’s nucleus-to-cytoplasm relocation” should be changed to “OTULIN removed M1 chains from cytoplasmic RTA, preventing its nuclear import”.

Response: Thank you very much for your insightful suggestion. We have revised the abstract to “OTULIN removed M1 chains from cytoplasmic RTA, preventing its nuclear import” according to your suggestion. We agree with you that this revised description is more accurate and appropriate for our working model on RTA regulation by LUBAC and OTULIN based on the current findings described in our manuscript.

REVIEWERS' COMMENTS

Reviewer #1 (Remarks to the Author):

The revised Luan et al. study addresses critical question in the field, by dissecting modification of a key viral protein. The authors have adequately addressed all the issues identified in the last review round by the reviewer. Additional information, including correction of the errors and requested quantification and statistical analysis was provided, as requested. The revised manuscript has been further improved. No further request for additional information.

Reviewer #2 (Remarks to the Author):

The revised manuscript has carefully addressed all of my comments.